# “The Loss of Golden Touch”: Mitochondria-Organelle Interactions, Metabolism, and Cancer

**DOI:** 10.3390/cells9112519

**Published:** 2020-11-21

**Authors:** Matteo Audano, Silvia Pedretti, Simona Ligorio, Maurizio Crestani, Donatella Caruso, Emma De Fabiani, Nico Mitro

**Affiliations:** DiSFeB, Dipartimento di Scienze Farmacologiche e Biomolecolari, Università degli Studi di Milano, 20133 Milano, Italy; Matteo.audano@unimi.it (M.A.); silvia.pedretti@unimi.it (S.P.); simona.ligorio@unimi.it (S.L.); maurizio.crestani@unimi.it (M.C.); donatella.caruso@unimi.it (D.C.)

**Keywords:** mitochondria, cancer, metabolism, subcellular organelles

## Abstract

Mitochondria represent the energy hub of cells and their function is under the constant influence of their tethering with other subcellular organelles. Mitochondria interact with the endoplasmic reticulum, lysosomes, cytoskeleton, peroxisomes, and nucleus in several ways, ranging from signal transduction, vesicle transport, and membrane contact sites, to regulate energy metabolism, biosynthetic processes, apoptosis, and cell turnover. Tumorigenesis is often associated with mitochondrial dysfunction, which could likely be the result of an altered interaction with different cell organelles or structures. The purpose of the present review is to provide an updated overview of the links between inter-organellar communications and interactions and metabolism in cancer cells, with a focus on mitochondria. The very recent publication of several reviews on these aspects testifies the great interest in the area. Here, we aim at (1) summarizing recent evidence supporting that the metabolic rewiring and adaptation observed in tumors deeply affect organelle dynamics and cellular functions and vice versa; (2) discussing insights on the underlying mechanisms, when available; and (3) critically presenting the gaps in the field that need to be filled, for a comprehensive understanding of tumor cells’ biology. Chemo-resistance and druggable vulnerabilities of cancer cells related to the aspects mentioned above is also outlined.

## 1. Overview of Cancer Cell Metabolism 

Proliferation, epithelial-mesenchymal transition, migration, and invasion are hallmark features of cancer cells; all these characteristics are obviously intertwined with cellular metabolism. In the following subsections, we review some key aspects of tumor metabolism.

### 1.1. ATP Demand in Cancer Cells

Although the metabolic phenotype might vary among different cancer cells, a core peculiarity is observed in tumors, as well as in normal proliferating cells, which is, the balance between ATP synthesis and biomass production [1].

Cell division and migration rely on cytoskeleton and organelles’ dynamic rearrangements. Thus, dividing and migrating cells depend on the ATP and GTP availability required to sustain actin-myosin microfilaments’ and microtubules’ reorganization, respectively. On the other hand, according to the “supply-based” model, the synthesis of the building blocks necessary for cell proliferation could be viewed as a strategy to decrease the high ATP:ADP ratio resulting from growth factor signaling and nutrient availability in cancer cells [2]. It follows that the metabolic reprogramming triggered by proto-oncogenes or mutated onco-suppressors, contributes to setting conditions favoring proliferation and biomass production. In this scenario, mitochondria play a central role because of their anabolic functions, mainly utilization of glutamine for lipid and nucleotide synthesis, rather than for their capacity to produce ATP via oxidative phosphorylation (Oxphos) [2].

Almost a century ago, Otto Warburg reported the observation that cancer cells consume more glucose than their “normal” counterparts and produce lactate even in the presence of oxygen, a phenomenon thereafter termed “aerobic glycolysis” or the “Warburg effect” [3]. This trait, which is typical of proliferating cells and not exclusively to cancer cells, could either be due to mitochondrial defects causing impaired oxidative metabolism or due to “abnormal” activation of glycolysis. Although some tumors are indeed characterized by oncogenic mutations in mitochondrial metabolic enzymes, it is now well-established that aerobic glycolysis is a direct consequence of most oncogenic mutations; the following metabolic rewiring predisposes cells to proliferation and transformation, independent of mitochondrial defects [4].

In this context, oxygen-independent activation of the Hypoxia Inducible Factor 1α signaling pathway is responsible for triggering the switch from oxidative metabolism to enhanced glycolysis, and conversion of pyruvate to lactate [5]. This is accomplished by inducing transcriptional activation of several metabolic enzymes such as, hexokinase 2; lactate dehydrogenase A, the isoform that more efficiently converts pyruvate to lactate; and pyruvate dehydrogenase kinase, the enzyme, which by inhibiting pyruvate dehydrogenase, impairs the mitochondrial metabolism of the pyruvate itself. It is worth noting that lactate should be considered much more than a metabolic substrate since it also exhibits paracrine effects on tumor-associated macrophages and effector T cells, promoting an “immuno-tolerant” environment that favors tumor cells’ growth [6].

Readers interested in a detailed discussion of the Warburg effect and its metabolic consequences could refer to milestone reviews, such as [1,7].

### 1.2. Metabolic Intermediates and Protein Modifications in Cancer Cells

To fully appreciate the impact of metabolism in tumors, another aspect should be kept in mind. Both abnormal activation of normal metabolic pathway, as well as defective or neomorphic enzymatic activities due to oncogenic mutations, provide a specific set of “signaling metabolites”. The increased availability of glutamine-derived acetylCoA moieties outside the mitochondria fuels fatty acid synthesis, mevalonate pathway, and protein acetylation. In fact, ATP-citrate lyase (ACLY), the enzyme that splits mitochondria-originated citrate into acetylCoA and α-ketoglutarate, was long recognized for its role in cancer and as a therapeutic target [8]. Indeed, ACLY has a dual role—(i) it generates the building block for lipid (fatty acids and cholesterol) synthesis linked to cell proliferation, and (ii) provides acetylCoA and isoprenoid intermediates for post-translational modification of proteins.

Lysine N-ε-acetylation is more commonly associated with histone proteins and transcription factors and regulators, contributing to the so-called histone code and epigenetic control [9]. In addition, a variety of non-histone proteins undergo acetylation, such as α-tubulin, which is a component of cytoskeleton microtubules.

The relationship between histone modifications, including acetylation, and certain types of cancers was long recognized and represents a promising pharmacological target in oncology. A thorough discussion of these aspects is far beyond the scope of the present review. Readers interested in the topic might refer to the published material, among which is the recent review by Neganova et al. [10]. 

Microtubule acetylation is a typical post-translational modification of stable and long-lived microtubules that enhances microtubule flexibility, and therefore, is resistant to mechanical stresses [11]. As mentioned above, cytoskeleton, and especially microtubules, are involved in proliferation, and especially in tumor invasiveness. In this regard, it is known that the more invasive cancer cells exhibit a higher frequency of long and dynamic microtubule-based membrane protrusions, with respect to the less invasive cell lines [12]. In this context, it was shown that acetylated α-tubulin levels are tightly correlated with aggressive metastatic behavior in breast cancer [13]. This representative article underlines the strict link between the cytoskeleton protein acetylation and cancer biology.

Isoprenoid units, i.e., geranyl and farnesyl groups, function as lipid anchors, favoring the attachment of modified proteins to the cell membrane. The Ras superfamily and the Rho family of GTPases are among the best-characterized prenylated signaling proteins with a direct involvement in cancer. Their role and the links with the mevalonate pathway in cancer biology were recently reviewed [14]. Here, we only mention a few cases exemplifying that indeed, these links and the multifaceted roles of prenylated oncoproteins in tumorigenesis and tumor progression are highly complex. For example, in models of lung cancers, mutated Ras proteins present cell transforming functions, regardless of their prenylation state, however, farnesylation mediated by farnesyl-transferase is required in later stages of tumor progression. Mutant p53 regulates the mevalonate pathway, by interacting with the lipogenic transcription factors Sterol regulatory element-binding proteins (SREBPs) in human cancers. On the other hand, the mevalonate pathway, by affecting the prenylation of regulatory proteins, stabilizes the mutant p53. Thus, inhibition of isoprenoid synthesis or specific inhibition of the activity of prenylating enzymes, represents a promising therapeutic approach.

### 1.3. Oncometabolites

In some cases, due to the mutations in metabolic enzymes, cancer cells accumulate specific intermediates, termed oncometabolites, exhibiting signaling properties. To date, the most characterized oncometabolites are—D-2-hydroxy-glutarate, succinate, and fumarate [15,16]. Abnormal accumulation of fumarate and succinate is found in cancer cells with loss-of-function mutations in the genes encoding the enzymes fumarate hydratase and succinate dehydrogenase, both belonging to the Krebs cycle. On the other hand, D–2-hydroxyglutarate results from mutated isocitrate dehydrogenase 1 and 2 (IDH1 and 2), due to the acquisition of neomorphic enzymatic activity. These metabolites might contribute to dysregulating cellular processes, ultimately promoting tumorigenesis. Readers interested in the specific signaling roles of these oncometabolites are referred to the milestone review by Yang et al. [15] and to the more recent review by Sciacovelli and Frezza [16]. In brief, one of the most relevant action of oncometabolites is the inhibition of α-ketoglutarate-dependent dioxygenases, leading to pseudohypoxia, epigenetic changes, such as DNA and histones hypermethylation.

### 1.4. Metabolic Heterogeneity in Cancer Cells

As mentioned above, all tumor cells are distinguished by growth factors-stimulated anabolic reactions and a favorable ATP:ADP ratio. Nevertheless, cancer cells are heterogeneous in terms of balance among different metabolic pathways, e.g., the Warburg effect vs. oxidative metabolism, glycolysis vs. the pentose phosphate pathway. Heterogeneity also relates to specific cellular functions, e.g., proliferation vs. migration, apoptosis vs. autophagy. Metabolic and functional heterogeneity depends on several factors, among which are the type of tumor, microenvironment, in the case of solid tumors, crosstalk with neighboring cells, such as stromal cells, immune cells, adipocytes, etc.

Mitochondria play a well-established role in cancer for several reasons—they are the site of key metabolic pathways, especially the Krebs cycle and glutamine metabolism; their structural plasticity and interactions with other organelles and cellular structures are crucial in cellular functions such as division and migration; they orchestrate metabolic substrate fueling and recycling by means of interaction with cellular compartment and organelles, e.g., cytoplasm, lipid droplets, autophagosomes; they operate as hubs for specific functions such as apoptosis, calcium ions trafficking, and iron homeostasis. Therefore, it is not surprising that the pharmacological targeting of mitochondria in cancer cells is considered a promising frontier in anti-cancer therapy [17].

Of note, mitochondria also play a role in other cell types and functions strictly related to cancer, such as endothelial cells during angiogenesis. For example, it was recently reported that inhibition of the respiratory chain complex III impairs proliferation, but not migration of the endothelial cells in vitro, by decreasing the NAD+/NADH ratio [18].

## 2. Fusion and Fission Event in the Onset and Progression of Cancer

Great interest is converging on mitochondria morphological remodeling, in response to internal and external cues, and the consequences of their rearrangement on energy metabolism in physiological and pathological conditions. Fusion and fission mechanisms are characterized by the condensation of two smaller mitochondria or the fragmentation of a single big mitochondrion, respectively [19]. It is now well-established that mitochondria modulate their morphology when cells are subjected to physical [20,21,22], chemical [23,24,25], and biological [26,27,28,29] cues.

In this context, controlled fusion-fission events are necessary to establish appropriate mitochondria number, shape, and localization [30,31]. For instance, cell proliferation is characterized by increased mitochondria fragmentation, to guarantee an equal distribution of mitochondria to the daughter cells [32]. In line with this notion, it was shown that glycolysis is increased upon mitochondria fragmentation, which is a peculiar trait of various cancer cells [33]. At the molecular level, fusion and fission events are tuned by well-characterized players such as, Optic Atrophy 1 (OPA1), involved in the inner mitochondrial membrane fusion; Mitofusin 1 and 2 (MFN1-2), which control the outer mitochondrial membrane fusion; Dynamin-1-like protein (DNM1L/DRP1) and Mitochondrial fission 1 protein (FIS1), both involved in the regulation of mitochondria fission [34]. Recent findings also helped identify new regulators of mitochondrial dynamics, shedding light upon new processes affected by mito-dynamics [35,36].

In addition, the physical and functional interaction between mitochondria and lysosomes was demonstrated to play a crucial role in mitochondrial function, indicating that autophagy is fundamental for a proper metabolic homeostasis, in both physiological and pathological conditions [37,38,39].

In this section, we specifically focus on how mitochondria dynamics and autophagy relate to intracellular metabolic status and how these interactions affect cancer onset/progression.

### 2.1. Mitofusin 1 and 2, Outer Mitochondrial Membrane Fusion, and Cancer

Elongation and condensation of mitochondria is a two-step event that requires the cooperation of the mitofusins (MFNs) family proteins and of OPA1, for the outer and inner mitochondrial membrane fusion, respectively. MFNs and OPA1 belong to the family of small GTPases, which use the energy released from GTP hydrolysis to generate membrane-membrane contacts and complete organelle fusion. Overall, alteration of MFNs or OPA1 function leads to decreased mitochondrial fusion, shifting the balance of mito-dynamics to over-fragmentation. This effect was observed in many experimental settings, aimed to investigate cancer biology. For instance, Zhang et al. demonstrated that MFN1 loss-of-function triggered the epithelial-to-mesenchymal transition of hepatocellular carcinoma (HCC) favoring HCC metastasis and invasiveness [40]. Of note, in vitro experiments indicated that MFN1 depletion reprogramed glucose metabolism, eliciting aerobic glycolysis instead of Oxphos. Contrasting results were collected by Li et al., who observed excessive mitochondrial fusion in the tumor tissues of human HCC and in vitro cultured tumor organoids from cholangiocarcinoma [41]. The knockdown of both MFN1 and OPA1 inhibited mitochondrial fusion in both experimental settings, leading to a reduced cell growth and tumor formation. The authors ascribed the antitumor effect of OPA1 and MFN1 silencing to the induction of pro-apoptotic mechanisms, inhibition of oxidative metabolism, and ATP production [41]. A realistic speculation might be that the apparently inconsistent observations could be due to the different experimental settings used in these studies.

Consistently with Zhang et al., it was shown that decreased levels of MFN1 mRNA were associated with ovarian cancer resistance to cisplatin [42]. Specifically, MFN1 expression levels were downregulated in SKOV3 and PA1 ovarian cancer cell lines, upon hypoxia, which caused reactive oxygen species (ROS) production, leading to an increased fission-to-fusion ratio of mitochondria. Strikingly, mitochondrial fission inhibition by Mdivi-1 treatment prevented resistance to cisplatin in both cell lines [42].

MFN2 is the paralog of MFN1, even if it was demonstrated that their efficiency in promoting mitochondrial fusion was significantly different [43]. In fact, MFN1 mitochondria are tethered in a GTP-dependent manner, more efficiently than MFN2 ones. Despite its lower pro-fusion capacity compared to MFN1, MFN2, together with Sirtuin1 (SIRT1), was demonstrated to mediate Yes-associated protein (YAP) activity in gastric tumor [44]. In vitro experiments in GES-1 and AGS cell lines indicated that the MFN2/SIRT1 axis contributed to boost mitophagy and the inhibition of caspase-9 mediated apoptosis. Of note, the authors also demonstrated that MFN2/SIRT1-promoted mitophagy led to lamellipodia formation, facilitating cell migration [44]. Pancreatic cancer seems to be one of the most affected by MFN2 activity. Indeed, pancreatic cancer cells expressing MFN2 showed decreased proliferation and ROS production, together with increased expression levels of the autophagy regulators LC3-II and Bax. In addition, MFN2 decreased the levels of phospho-phosphoinositide 3-kinase (p-PI3K), phospho-protein kinase B (p-Akt), and phospho-mammalian target of rapamycin (p-mTOR) proteins, which promote pancreatic cancer cell growth, proliferation, and metastasis [45]. Contrasting results were collected by Pan et al. who demonstrated that miR-125a by disrupting the MFN2 expression, favored mitochondrial fission, apoptosis, metabolic rearrangement in the mitochondria, and impairment of cell migration into pancreatic cells [46]. Specifically, low levels of miR-125a upregulated MFN2 expression in the human epithelioid carcinoma cell line PANC-1, favoring Oxphos activity and mitochondrial fusion, ultimately leading to cell survival and migration [46]. Recent findings also found a significant correlation between bladder cancer and MFN2 activity; MFN2 was significantly downregulated in bladder cancer patients and its expression was associated with the tumor stage [47]. In line with these findings, MFN2 silencing promoted cell migration, proliferation, and invasion, and increased tumor progression both in vivo and in vitro [47].

### 2.2. OPA1 and the Role of Inner Mitochondrial Membrane in Cancer

OPA1 is directly involved in the fusion events of the inner mitochondrial membrane, maintenance of mitochondrial DNA (mtDNA), and cristae formation [48]. Evidence showed that at least eight isoforms of OPA1 exist in eukaryotes, where every isoform has specific functions. Overall, “long” isoforms determine mitochondrial fusion, whereas “short” ones are involved in mitochondrial fission [49,50]. Given the crucial role of OPA1 in the regulation of mitochondrial morphology and function, several studies focused on the relevance of its activity in cancer [51,52]. Recent research work demonstrated increased OPA1 levels in ovarian cancer, together with increased mitochondrial oxidative metabolism, mitochondrial number, expression of the two master regulators of mitochondrial biogenesis, peroxisome proliferator-activated receptor gamma coactivator 1-alpha (PGC1α), and mitochondrial transcription factor A (TFAM) [53]. Moreover, OPA1 mediated, at least in part, the antitumoral activity of Withaferin A in breast cancer cells, a steroidal lactone derived from the medicinal plant *Withania somnifera* (WA) [54]. The treatment of the human breast cancer cell lines MCF-7 and SUM159 with WA, showed a reduced assembly and expression of cytochrome reductase (complex III of the electron transport chain), MFN1 and 2 expression levels, mitochondrial volume, and fusion [54]. Noteworthy, Herkenne S. et al. demonstrated for the first time that OPA1 is also required for developmental and tumor angiogenesis [55]. In vascular endothelial cells, OPA1 expression responded to angiogenic stimuli and subsequently inhibited nuclear factor kappa-light-chain-enhancer of the activated B cell (NF-κB) signaling pathway. This ultimately favored the expression of pro-angiogenic genes and angiogenesis, facilitating tumor formation and invasiveness [55].

OPA1 was also lately linked to nervous system tumors. In fact, chromosome 3q26 gene products SRY-Box Transcription Factor 2 (SOX2) and OPA1 were found to be differentially regulated in invasive gliomas [56]. The authors well-characterized a mutually exclusive mechanism in which SOX2 blunted OPA1 expression levels in this type of cancer. Strikingly, they found that OPA1 inactivation increased LN319 glioma cell invasion in vitro, and boosted cell dispersion in xenotransplanted Danio rerio embryos [56]. Ando et al. also evidenced the relevance of Kinesin Family Member 1B (KIF1Bβ)/YME1 Like 1 ATPase (YME1L1)/OPA1 axis in neuroblastoma prognosis [57]. As discussed by the authors, KIF1Bβ, which is frequently deleted in neuroblastoma, physiologically interacts with YME1L1, a mitochondrial protease, favoring OPA1 cleavage and mitochondrial fission. Overexpression of YME1L1 favored mitochondrial fragmentation, and ameliorated the prognosis of neuroblastoma, inducing the apoptosis of in vitro neuroblastoma cell lines [57].

Taken together, these studies describe a crucial role of mitochondrial fusion proteins in cancer, indicating a key role of these regulators in tuning the energetic homeostasis of tumor cells.

### 2.3. DRP1 and Mitochondrial Fission in Cancer Biology and Metabolic Pathways

Similar to the MFNs and OPA1, DRP1 is a GTPase protein that belongs to the dynamin proteins superfamily. In humans, there are six different isoforms of DRP1, including a brain-specific one. DRP1 isoforms show different tissue expression patterns and peculiar features related to mitochondrial fission [58,59]. Since mitochondrial fission is widely recognized to be directly associated with tumorigenic phenotypes, in the last couple of years, the role of DRP1 in cancer biology was extensively investigated in different types of tumors, such as pancreatic, breast, colorectal, lung, liver, prostate, and ovarian cancer. Moreover, the function of DRP1 was also studied in immune cells, investigating the process through which mitochondrial fission might impact the relationship between tumor and immunogenic response.

Concerning pancreatic cancer, it was shown that DRP1 was significantly upregulated in pancreatic cancer cell lines and tissue samples [60]. As stated by the authors, DRP1 promoted cell growth and invasiveness, both in vitro and in vivo, promoting G1-S phase progression and metalloproteinase-2 expression. In addition, DRP1 concomitantly increased glycolysis activity to mitochondrial fission [60]. In line with these findings, a study by Nagdas et al. indicated that DRP1 was required for the KRas-driven anchorage-independent growth in fibroblasts and patient-derived pancreatic cancer cell lines, promoting glycolysis flux [61]. Nevertheless, the authors also demonstrated that DRP1 deletion in pancreatic cancer cells conferred a significant survival advantage in a model of KRas-driven tumor. In fact, tumor cells lacking DRP1 showed a normal glycolytic flux, together with impaired mitochondrial morphology, ultimately leading to impaired TCA cycle and fatty acid β-oxidation [61]. Collectively, these findings support the role of DRP1 as the metabolic mediator of glycolysis and mitochondrial function, by promoting mitochondria fusion/fission cycling and mitophagy.

Consistent results were obtained in the KRas mutant non-small-cell lung cancer (NSCLC), where DRP1 orchestrated a metabolic rewiring to promote lactate utilization and ROS production suppression [62]. In the same type of tumor, DRP1 was involved in metabolic rearrangements that favored resistance to chemotherapy [63]. The authors identified a molecular axis represented by Proto-Oncogene Serine/Threonine-Protein Kinase (PIM1), DRP1, and mitochondrial fission; PIM1 inhibition favored DRP1 activity and mitochondrial fragmentation, increasing mitochondrial ROS production and resistance to pharmacological therapy [63]. Further studies suggested an antitumoral property of DRP1 in other types of lung cancer. For instance, Guo and colleagues showed that DRP1 is repressed by LIM Zinc Finger Domain Containing 1 (PINCH-1) in lung adenocarcinoma [64]. In this context, PINCH-1 inhibition led to increased DRP1 activity, promoting mitochondrial fission and blunting the expression of Pyrroline-5-Carboxylate Reductase 1. This mechanism resulted in the significant inhibition of proline synthesis and cell proliferation [64].

An antitumoral activity of DRP1 was also proposed by Zhang et al. through in vitro studies in a model of liver cancer [65]. Researchers showed that Large Tumor Suppressor 2 (LATS2) overexpression increased DRP1 protein levels, leading to aberrant mitochondrial fragmentation and ultimately to mitochondrial dysfunction, characterized by a mitochondrial membrane potential reduction, mitochondrial respiratory complex downregulation, mitochondrial cytochrome-C (Cyt-C) release into the nucleus, and consequential apoptosis [65]. Consistently, through a tissue array analysis, Young Yeon Kim et al. demonstrated that DRP1 was significantly decreased in malignant colon and lung cancer tissues, whereas no changes in the DRP1 levels were observed in breast and prostate tumors [66]. Further analyses by the same group proved that DRP1 levels were downregulated in both advanced-grade colon and lung cancers, suggesting that DRP1 loss is associated with the progression of human lung and colon cancer. Nevertheless, recent studies indicated that DRP1 might be involved in both the androgen-induced metabolic reprogramming of mitochondria in prostate cancer (PC) cells [67] and colorectal cancer (CRC) [68]. DRP1 expression levels were increased in androgen-sensitive and castration-resistant, androgen receptor-driven PC, where it favored the formation of voltage-dependent anion channels (VDAC)—mitochondrial pyruvate carrier 2 (MPC2) complex; this event boosted pyruvate transport into mitochondria, Oxphos, and lipid synthesis, supporting cell growth and proliferation [67].

Similar effects were observed in the colorectal cancer studies, where DRP1 activity upregulation by high-mobility group box 1 protein (HMGB1)/receptor for advanced glycation end product (RAGE)/ERK axis, increased cell growth and chemoresistance [68]. Strikingly, the authors indicated that inhibition of DRP1 phosphorylation at Ser616 (DRP1^Ser616^) by HMGB1 and RAGE inhibitors, significantly improved sensitivity to the chemotherapeutic treatment, by suppressing autophagy [68]. DRP1^Ser616^ was also implicated in chloroquine- and isorhamnetin-mediated inhibition of triple-negative breast cancer cell growth and proliferation [69]. Specifically, CaMKII phosphorylation at Thr286 (CaMKII^Thr286^) and DRP1^Ser616^, promoted Bax translocation to mitochondria, Cyt-C release, and consequential cell apoptosis.

On the other hand, DRP1 seems to mediate cisplatin (CDDP) resistance in ovarian cancer, under hypoxic conditions [42]. Specifically, hypoxia-induced ROS decreased the inhibitory Ser637 phosphorylation of DRP1 (DRP1^Ser637^) and increased mitochondrial fission. Notably, inhibition of mitochondrial fission rescued CDDP sensitivity to hypoxic ovarian cancer cells [42]. Together, these findings suggest that researches aiming to identify new chemotherapeutics might specifically target DRP1 to overcome drug resistance in tumors.

Noteworthy, immune cell therapy is now proposed as a novel treatment for tumors [70,71]. In this perspective, it was demonstrated that DRP1 is required for an optimal anti-tumor response of T-cells [72]. The authors showed that DRP1 fostered (*i*) developing thymocytes migration and expansion, (*ii*) effector T-cell metabolic reprograming involving calcium/AMP-activated protein kinase (AMPK)/mTOR axis upon their activation, and (*iii*) migration and extravasation of T-cells, and finally avoided the shift toward a memory-like phenotype of T-cells in tumor environment. These data suggest that DRP1-engineered cells could represent a valuable tool for future cell-based therapies against cancer.

Another mitochondrial fission regulator is FIS1, whose expression and activity was recently linked to the cancer phenotype, by several reports. A research conducted by Abo Elwafa et al. indicated that FIS1 was significantly overexpressed in the bone marrow of acute myeloid leukemia (AML) patients, and that high FIS1 levels showed a significant negative impact on complete remission response after therapy [73]. Importantly, FIS1 overexpression in AML was independent of other variables, such sex or age. Consistently, Shanshan Pei et al. published a paper where they found that AMPK/FIS1-mediated mitophagy is required for self-renewal of human AML stem cells [74]. Human AML leukemia stem cells (LSCs) display high activity of the AMPK/FIS1 axis and its unique mitochondrial morphology characterized by hyper-fragmented organelles. This feature was blunted by FIS1 silencing, which also led to decreased mitophagy, inhibition of the tumorigenic protein GSK3, cell cycle arrest, and increased cell differentiation of LSCs in AML [74].

Another report showed that the JNK-FIS1 biological axis is important for mitochondrial stress mediated by Sirtuin 3 (SIRT3) inhibition in tongue cancer [75]. The authors used two types of tongue cancer cells (SCC9 and SCC15), in which they demonstrated that SIRT3 silencing led to mitochondrial oxidative stress, energy metabolism disorder, mitochondrial Cyt-C release, and mitochondrial apoptosis activation [75].

Overall, these research studies indicated that not only DRP1, but also FIS1 could be considered as a valuable target for future therapeutic approaches against cancer. The key mechanisms described in this section and the role of gene in cancer are summarized in Figure 1 and Table 1.

## 3. Inter-Organellar Communications in Cancer Cells

Mitochondria interact with different subcellular organelles and cell structures. In the following section, we focus on the changes that were more recently reported to occur during tumor development or progression between the mitochondria-endoplasmic reticulum, mitochondria-lysosome, and mitochondria-cytoskeleton interactions.

### 3.1. Mitochondria-Endoplasmic Reticulum and Mitochondrial-Associated Membranes

The first site of contact discovered between two intracellular organelles was the physical interaction between the mitochondria and endoplasmic reticulum (ER), originally described in 1959 by Copeland and Dalton, which observed a possible association between ER and mitochondria, in the cells of a pseudobranch gland of a teleost [76]. This pioneering discovery was further confirmed by another group, a decade later [77]. Only twenty years later, distinct structures called mitochondrial-associated ER membranes (MAMs) were purified from rat liver by Vance, who proposed that MAMs are important for phospholipids transfer between ER and mitochondria [78,79]. The same group, during the years, demonstrated that MAMs are involved in the regulation of lipid synthesis and transport. In this context, MAMs function as hubs where the enzymes of lipid synthesis and transport pathways are located, both at the endoplasmic reticulum and at the mitochondrial membranes [80,81].

Rizzuto et al., by means of two differently colored and specifically targeted fluorescent proteins, reported numerous close contacts between ER and mitochondria, to maintain calcium (Ca^2+^) homeostasis. These authors also estimated that the surface of the mitochondrial network in close apposition to the ER was ∼5 to 20% of the total [82].

Extracellular stimuli activate the release of Ca^2+^ from the ER through the action of 1,4,5-trisphosphate (IP3) and ryanodine receptors (IP3Rs, RyRs) and of the second intracellular messenger IP3. Then, VDAC, present on the outer mitochondrial membrane (OMM) at the contact sites between ER and mitochondria, mediates the transfer of different ions and molecules into the intermembrane space [82,83,84]. Differing from the OMM, the inner mitochondrial membrane (IMM) is not permeable to Ca^2+^, which is transported into the mitochondrial matrix through the mitochondrial calcium uniporter (MCU), a protein independently isolated by two different groups, almost ten years ago [85,86]. Ca^2+^ import into mitochondria is key for mitochondrial metabolism, while Ca^2+^ overload into mitochondria results in cell death [87,88].

Another key role of MAMs is their involvement in the regulation of mitochondrial bioenergetics, morphology, and motility—indeed the close juxtaposition with other organelles regulates the dynamic network of mitochondria. In this context, the mitochondrial Rho-like GTPase Miro directly interacts with Milton and Kinesin-1 to form a Ca^2+^-sensitive protein complex for the anterograde transport of mitochondria [89,90]. Moreover, Fun14 domain-containing 1 (FUNDC1) integrates mitochondrial fission and mitophagy at the interface of the MAM, by working in concert with DRP1 and calnexin, under hypoxic conditions in mammalian cells [91]. A role of MAMS in inflammation and ER stress was also reported [92,93]. 

Given the pleiotropic roles of the tethering between ER and mitochondria, and more specifically the implications of MAMs, it is not surprising that these platforms also play a role in carcinogenesis. Several studies reported MAMs to be involved in the transfer of Ca^2+^ and ROS, in response to functional alterations of oncogenes and oncosuppressors [94,95,96]. In the last few years, several reviews exhaustively described different aspects of MAMs involvement in cancer [97,98,99,100,101,102,103]. In the following paragraphs, we summarize the most relevant or recent evidence that emerged in the area.

It was reported that several tumors showed altered lipid composition of the IMM, which impacts on mitochondrial functions and ultimately on apoptotic cell death [104]. For instance, increased levels of mitochondrial cholesterol were detected in hepatocellular carcinoma [105,106]. Indeed, higher levels of cholesterol were linked to increased aerobic glycolysis, due to the negative effects on mitochondrial membrane potential and the altered activity of the Oxphos [107].

Moreover, the serine/threonine kinase Akt was overactivated in tumors characterized by increased cholesterol levels [108,109]. More recently, Mignard and coworkers demonstrated that ceramide, especially C16-ceramide levels increased during early apoptosis, possibly through a conversion from mitochondrial sphinganine and sphingomyelin, although sphingosine and lactosyl- and glycosyl-ceramide levels were unaffected. Moreover, ceramide generation was enhanced in mitochondria when the sphingomyelin levels were decreased in the MAMs. The authors concluded that significant sphingolipid modifications occur in MAMs, mitochondria, and ER, during the early steps of apoptosis, a key pathway escaped by cancer cells [110]. These studies uncover a specific role of MAMs, as the key sites where remodeling of the sphingolipid composition is functionally linked to apoptosis, with implications in cancer cells homeostasis.

ER-mitochondria contact sites are also a primary platform for decoding danger signals, such as variation in Ca^2+^ homeostasis, which can be perturbed by oncogenes and oncosuppressors to determine cancer development or progression [111]. MAMs play a key role in regulating Ca^2+^ flux from ER to mitochondria. Intracellular signaling altering Ca^2+^ transfer leads to Ca^2+^ overload in the mitochondrial matrix and activation of the permeability transition pore (PTP). These events drive the increase in permeability of IMM and the impairment of mitochondrial membrane potential, leading to mitochondrial swelling, rupture of OMM, release of cytochrome c, and apoptosis [100].

The activity of the IP3Rs enriched at the ER-mitochondria interface is under the control of oncogenes, which by phosphorylation, regulate Ca^2+^ homeostasis [112]. As mentioned above, in cancer cells, Akt is overactivated, therefore it primarily hyperphosphorylates IP3R3 (the main isoform of IP3Rs present in MAMs), which leads to reduced Ca^2+^ release from ER to mitochondria, ultimately favoring cell survival instead of apoptosis [113]. Furthermore, the localization of the mammalian Target of the Rapamycin Complex 2 (mTORC2) at MAMs has a dual role—(*i*) it activates Akt to favor cell survival and (*ii*) it phosphorylates hexokinase 2 to promote the Warburg effect [114,115]. Recently, Ciscato et al. demonstrated that displacement of hexokinase 2 from MAMs, with a selective peptide, favors mitochondrial Ca^2+^ overload, resulting in mitochondrial depolarization and cell death [116].

In addition, the tumor suppressor promyelocytic leukemia (PML) protein localizes at MAMs. Here, PML is part of complex including IP3R3, Akt, and protein phosphatase 2A (PP2A). This complex allows the dephosphorylation and consequently the inactivation of Akt by PP2A, which therefore resulted in reduced IP3R3 phosphorylation. All of this leads to the maintenance of susceptibility to the Ca^2+^ levels to produce an apoptotic signal [117].

Another fundamental tumor suppressor gene such as p53, also localizes at MAMs. In this context, p53 interacts with the sarco/endoplasmic reticulum Ca^2+^ ATPase (SERCA) pump to fill Ca^2+^ storage in ER. An apoptotic signal or treatment with antineoplastic agents favor Ca^2+^ flux from ER to mitochondria, enabling apoptosis [118]. On the other hand, thioredoxin-related transmembrane protein 1 (TMX1), by inhibiting the SERCA pump at MAMs and reducing ER Ca^2+^ flux, impairs mitochondrial function. Interestingly, cancer cells show lower levels of TMX1 with increased Ca^2+^ content in the ER, which however, displays a reduced ability to flux Ca^2+^ to mitochondria [119].

Given that the MCU complex plays an essential role in mitochondrial Ca^2+^ uptake for many cellular functions, its role in tumorigenesis has started to emerge in several studies [120]. MCU is overexpressed in several cancers such as colorectal, ovarian, prostate, and breast cancer, and is associated with genetic alterations as gene amplifications. miR-25 was identified as MCU-targeting microRNA. Indeed, its overexpression reduces the expression of MCU, mitochondrial Ca^2+^ uptake, and the resistance of cancer cells to apoptotic challenges, thus, favoring tumor cell survival [121]. Latest studies implicate mitochondrial Ca^2+^ dynamics in cell migration and a metastasis promoting role for the MCU-dependent mitochondrial Ca^2+^ uptake in different tumors [122,123].

In cancer, alterations in mitochondrial morphology and function also results in dramatic changes on the whole cell metabolism. As mentioned above, the Warburg effect consists of the upregulation of glycolysis and a concomitant reduced flux of pyruvate into mitochondria, which is converted to lactate. All this provides glycolytic ATP production and cancer cell growth advantage. In the last few years, a role for mitochondria in cancer was reconsidered, since several intermediates of the tricarboxylic acid (TCA) cycle, an amphibolic pathway, used in the synthesis of lipids, nucleotides, and proteins; the availability of these “building blocks”, coupled with a favorable ATP:ADP ratio, support the high rate of proliferation of cancer cells. To maintain oxidative activity, cancer cells adapt to the use of other fuel sources other than glucose, such as glutamate or fatty acids [124]. Accordingly, the reduction of mitochondrial Ca^2+^ uptake, as a consequence of IP3Rs or MCU activity inhibition, results in decreased intracellular ATP levels, which in turn activate the AMPK-dependent autophagy [125]. Of note, this effect was linked to the deficient Ca^2+^-mediated stimulation of the TCA cycle [126].

Another key mechanism important in the regulation of cell survival is the control of the distance between ER and mitochondria. Indeed, using synthetic linkers to favor the tightening between ER and mitochondria leads to mitochondrial Ca^2+^ overload and apoptosis. Conversely, loosening without disrupting these ER-mitochondria contacts sites stimulates mitochondrial respiration and ATP production [127].

Trichoplein/Mitostatin (TpM) is a protein that regulates ER-mitochondria distance. TpM is a negative regulator of the tethering of ER with mitochondria, thus, it inhibits apoptosis by Ca^2+^-dependent stimuli, through interaction with MFN2 in MAMs [128]. TpM was also reported to be downregulated in the advanced stage of human prostate cancers, suggesting that it might function as a tumor suppressor; thus, the measure of its expression might represent a useful clinical marker for diagnosis and prognosis in this kind of cancer [129].

A recent study revealed that Nogo-B/Reticulon is overexpressed in hepatocellular carcinoma and it has an oncogenic role in the progression of this tumor [82]. Nogo-B/Reticulon is another protein that regulates ER-mitochondria distance. Indeed, this protein is induced by hypoxia, leading to increased distance between ER-mitochondria, which in turn results in alteration of both phospholipid and Ca^2 +^ transfer, as well as of the apoptotic pathway [130]. This protein is up-regulated in human colorectal cancer and by interacting with the antiapoptotic protein Caspase 8 (CASP8) and FADD like apoptosis regulator (CFLAR, also known as c-FLIP), it plays a key negative role in apoptosis [131].

FATE1 (Fetal and Adult Testis Expressed 1) is another protein localized at the interface between ER and mitochondria, and is enriched in MAMs. Of note, FATE1 expression is low in normal adrenal cortex and in benign tumors (hyperplasia and adenoma), but is overexpressed in adrenocortical carcinoma. Given the above, physiologically, FATE1 decreases mitochondrial Ca^2+^ uptake, while its overexpression increases the resistance of adrenocortical cells to apoptosis, even if induced by mitotane treatment, a drug commonly used in the therapy of advanced adrenocortical carcinoma [132]. In this regard, a previous synthetic lethal screen study unraveled that loss of function of FATE1 was able to sensitize different non-small cell lung cancer cell lines to paclitaxel toxicity [133]. Moreover, Maxfield and coworkers showed that FATE1 inhibits proapoptotic signaling in different cancer cell lines, by destabilizing the pro-apoptotic BCL2 interacting killer (BIK) protein [89]. Collectively these data highlight that FATE1 plays a key role in the aggressive phenotype of adrenocortical carcinoma and is also implicated in the resistance to chemotherapeutic treatments in other kinds of cancer.

Recently, it was reported that Galectin-3, a β-galactoside-binding lectin, localizes at the ER-mitochondria interface to coordinate the functioning of the ER and mitochondria, preserves the integrity of mitochondrial network, and modulates the ER stress response. Specifically, Galactin-3 prevents the activation and recruitment at the mitochondria level of DRP-1 to avoid fission. Accordingly, loss of Galectin-3 impairs mitochondrial morphology and dynamics, leading to a more fragmented and rounded mitochondria, both in normal and cancer epithelial cells in basal conditions; on the other hand, depletion of Galectin-3 in pancreatic cancer cells was reported to be associated with increased Oxphos activity [134].

In summary, the evidence that cancer cells are addicted to constitutive ER-mitochondrial Ca^2+^ transfer, represents a promising therapeutic strategy to target tumor cells, by suppressing their survival and invasion. Some of the mechanisms described in this section are reported in Figure 2.

### 3.2. Actin Polymerization and Mitochondria-Cytoskeleton Interaction in Cancer Cells

The cytoskeleton is an interconnected network of filamentous polymers (microfilaments, actin filaments, and microtubules) that can be assembled or disassembled dependently on the cell’s requirements. These polymers are associated with regulatory proteins and work together to allow the main functions of the cytoskeleton, such as (*i*) organization of coordinated forces that allow cell movement and shape, (*ii*) spatial arrangement of cell content, and (*iii*) connection between cell and external environment [135]. The key features of cancer cells are rapid and uncontrolled growth, changed morphology, and migration. In these processes, cytoskeleton remodeling plays an important role [136]. External signals are received from mechanosensors and translated into internal responses by the cytoskeleton, activating different pathways such as the Hippo pathway, the focal adhesion kinases (FAK), GTP-binding protein RhoA, and JAK/STAT [137].

In a study conducted by Padilla-Rodriguez et al., the role of estrogen against breast cancer dissemination and metastasis was investigated. More specifically, the authors found that the estrogen receptor (ER) is involved in actin reorganization, which in turn is implicated in cancer cell invasion, the initial step in metastatic dissemination [138]. Using an in vitro system, they showed that ER inhibition enhanced MCF7 cell invasion and decreased proliferation, while estradiol (E2) treatment suppressed invasion and promoted proliferation. Furthermore, the authors investigated the motility dynamics of membrane protrusion as a marker of cancer cell invasive potential. Enhanced motility is associated with aggressive invasive behavior, and the actin cytoskeleton regulates this process. E2-treated cells presented the cortical actin bundle and decreased levels of protrusions (suppressive cortical actin bundles, SCABs). On the contrary, ER inhibition enhanced dissemination through protrusion formation and decreased the presence of actin bundles. Following inhibition of cell contractility by means of the Rho-associated protein kinase (ROCK) inhibitor (Y-27632), the authors found that ROCK inhibition provoked the decrease in cortical actin bundles, and this process was linked with protrusive activity. The hypothesis was that ER-induced actin cytoskeleton remodeling is a potential target for invasively inhibiting breast cancer and metastasis. The most differentially expressed actin cytoskeleton regulator, EVL, was induced by E2 treatment and promoted ER-mediated actin remodeling. Interestingly, the EVL-depleted cells showed unresponsive behavior to E2 treatment and a lack of SCABs generation. On the other hand, overexpression of eGFP-EVL restored SCABs formation and limited protrusion capacity [138]. 

In another work, Da Silva et al. studied the interaction between breast cancer cells metastasis, cytoskeleton rearrangement, and prolactin (PRL) [139]. They investigated actin remodeling after PRL treatment in breast cancer cells, and they found an increase in ruffles and pseudopodia formation and thickness of the perimembrane area, indicating an increased migration activity. PRL regulated cytoskeletal controllers, moesin, and focal adhesion kinase (FAK) through c-Src, activated by auto-phosphorylation, after interaction with the cell-membrane receptors. Both moesin and FAK are overexpressed in breast cancer, and their expression is linked to cytoskeleton remodeling, leading to cell motility, and migration potential [139]. The described mechanisms are summarized in Figure 3.

Another interesting aspect of cytoskeleton reorganization recently emerged in cancer research—actin remodeling to avoid Natural Killer (NK)-cell-mediated cell lysis. NK cells are innate the immune system’s lymphocytes with cytotoxic activity and are particularly important in recognizing and destroying cancer cells. Immunologic synapse (IS) formation between immune and target/tumor cells is a key process in the NK-mediated cell death. Actin filaments (AFs) network is in the center of IS and controls the delivery of myosin IIA-associated granules and, consequentially, that of cell lysis. 

Based on these premises, Absi et al. reported that actin cytoskeleton remodeling protected breast cancer cells from NK-mediate cell lysis [140]. They used two different types of cancer cells—the MDA-MB-231 cell line, which is highly resistant to NK attack, and MCF-7 cell line, which is extremely susceptible. The cells were placed in contact with the NK, and then the NK-cell-conjugated MDA-MB-231 cells presented the so-called “actin response”, showing intense accumulation of actin near the IS, whereas, the MCF-7 cells did not produce this reaction after contact with NK-cells. Hence, tumor cells capable of actin response remained alive after immune cell release, while those cells that could not rearrange their cytoskeleton were efficiently lysed by NK cells. To prove the key involvement of cytoskeleton rearrangement, the authors tested the inhibition of actin response (N-WASP or Cdc42 knockdown) in the MDA-MB-231 cell line and observed increased susceptibility to the immune cells’ attack.

The role of the cytoskeleton in the development and migration of cancer cells concerns breast cancer cells, and other types of tumor cells. For instance, Destrin, a member of the ADF/cofilin family, is associated with lung cancer tumorigenesis and malignancy [141]. Arp2/3 is involved in melanoma migration and progression [142], and activation of Cdc42 is studied as a potential therapeutic strategy in pancreatic ductal adenocarcinoma (PDAC) [143].

Additionally, the connection between the cytoskeleton and mitochondria is of great importance in patho-physiology. Indeed, F-actin is linked with mitochondria (mitochondrial F-actin) and is regulated by several actin regulatory proteins that control mitochondrial morphology and function.

In a recent study, this link between mitochondria and cytoskeleton was highlighted. The authors observed DRP1-independent mitochondrial fragmentation, following metabolic stress due to a lack of Ca^2+^ transfer. In this process, SIRT1 was activated and acted on cortactin, which induced actin polymerization and consequently mitochondrial fragmentation. The authors observed that the knockdown of cortactin or inhibition of actin polymerization prevented mitochondrial fragmentation induced by metabolic cellular stress, thus, SIRT1 emerged as a new player that acts by regulating the actin cytoskeleton [144].

The interaction between mitochondria and cytoskeleton was also studied in cancer cells known to exhibit adapted mitochondrial dynamics, to resist apoptosis and cope with energy demands. In fact, another mitochondrial process controlled by cancer cells is mitophagy, a cellular autophagy event leading to selective degradation of damaged or dysfunctional mitochondria. A fundamental prerequisite for mitophagy is mitochondrial fission. Since the cytoskeleton influences this process, Li et al. investigated the involvement of actin and cytoskeleton in the process of mitophagy in breast cancer cells [145]. Cofilin is an actin-binding protein involved in actin depolymerization and increased actin filament turnover. Overexpression of cofilin is associated with the aggressiveness of different types of cancer, but the exact mechanism is yet unknown [146]. The authors showed that mitochondrial fragmentation and mitophagy caused cofilin translocation from cytosol to mitochondria. Furthermore, knockdown of cofilin reduced mitochondrial fission and mitophagy through the indirect regulation of PINK1 (PTEN-induced kinase 1) in breast cancer cells treated with different compounds, which induce these mitochondrial processes [145].

Moreover, tumor cells can reposition their mitochondria from perinuclear localization to cortical cytoskeleton (mitochondrial trafficking), to provide energy to the membranes and enable membrane dynamics, thus promoting the formation of metastases. Syntaphilin (SNPH) is a molecule that is capable of inhibiting mitochondrial trafficking and is a regulator of tumor metastasis. In a study conducted by Seo et al., SNPH was shown to be ubiquitinated in tumor cells, and this ubiquitination at different lysine residues allowed the association with tubulin and inhibited mitochondrial movements. On the contrary, increased mitochondrial trafficking and mitochondria repositioning in the cytoskeleton cortical area, near the membranes, was observed when SNPH ubiquitination was decreased. All this resulted in tumor cell invasion [147].

Cytoskeleton remodeling is also involved in mitochondrial dynamics and mitochondrial trafficking, and controls the activity and consequently cellular metabolism in cancer cells. Lin et al. studied the role of fascin, an actin-bundling protein, in lung cancer metastasis. Fascin was highly expressed in lung cancer, and its overexpression was associated with increased tumor metastasis, while its depletion blocked the metastatic expansion of the disseminated lung cancer cells. The main effects of fascin depletion were on metastatic expansion and only had a marginal effect on tumor growth. This observation underlines that although cytoskeleton remodeling is involved in both cell proliferation and tumor invasion, specific proteins regulating actin filament assembly might impact only one of the two processes. Furthermore, the authors investigated the effect of fascin on cellular metabolism, and they showed that this protein controlled metabolic stress resistance and enhanced mitochondrial Oxphos activity. To confirm that fascin’s role was due to F-actin and mitochondrial interaction, they showed that mutations at the two actin-binding sites abolished the effect of the protein on mitochondria activity. In addition, they found that fascin-depleted lung cancer cells exhibited damaged mitochondrial F-actin, which caused mtDNA loss and inhibition of the respiratory complex biogenesis. As a consequence, fascin depletion blocked metastatic expansion [148] thus, confirming the requirement of proper F-actin-mitochondria interaction to support motility and invasion of metastatic cancer cells.

The studies discussed in the present section demonstrated how the regulation of the cytoskeleton and the interaction between actin and mitochondria could influence the behavior of cancer cells, and consequently opens up new perspectives for the identification of potential targets in cancer therapy.

## 4. Autophagosome: A New Player in Cancer Biology and Metabolism

Autophagy is a fundamental and highly conserved process, whose final outcome is the degradation and possible recycling of unnecessary or damaged material inside the cell. Its role in many physiological processes, such as development and differentiation was largely demonstrated [149,150,151,152,153]. Several studies helped to investigate the tight relationship between autophagy and mitochondria, highlighting a major role of this interaction in intracellular metabolic regulation [154,155]. Plus, a plethora of evidence showed the involvement of autophagosome, the cellular structure involved in the early steps of autophagy, also in cancer biology [156,157]. Together, these research lines paved the way to find a direct link between autophagy and metabolism, in the tumoral environment [158,159]. The central idea is that autophagy enables tumor cells to survive in high-stress conditions, such as metabolic or oxygen restriction, giving them an advantage, as compared to normal cells. To do this, autophagy imposes intracellular metabolic changes that guarantee tumor cell survival. More recent advances helped in elucidating this intricated connection.

In particular, Cho et al. showed that the natural compound Matrine, an alkaloid extracted from the herb *Sophora flavescens* that inhibits lysosomal proteases, inhibited cell growth of KRas-driven pancreatic cancer cells, by preventing autophagy dependent-metabolic changes, both in vivo and in vitro [160]. Strikingly, supplementation with the metabolic substrates α-ketoglutarate or pyruvate, rescued cell growth in Matrine-treated cells.

The impact of autophagy on energy metabolism was also described under hypoxia and glucose restriction, which significantly induce autophagic response [161,162]. The authors demonstrated that glucose or oxygen restriction deactivated anabolic pathways, increased hypoxia inducible factor-1α (HIF-1α), and autophagic machinery in IMR90 fibroblasts [25]. In particular, they observed increased levels of energy pathway intermediates belonging to the pentose phosphate pathway, glycolysis, TCA cycle, and amino acid metabolism, together with mitophagy activation. As stated by the authors, the results collected from this study were in line with previous observations obtained in cancer cells and contribute to understanding the relationship between autophagy and metabolism in tumoral settings [25].

Another study described how autophagy mediates the antitumoral effects of CPI-613, a lipoate analog that inhibits the mitochondrial enzymes pyruvate dehydrogenase (PDH) and α-ketoglutarate dehydrogenase, in clear cell sarcoma (CCS) [163]. In vitro experiments indicated that CPI-613 induced autophagosome formation, followed by lysosomal fusion in HS-MM CCS cells.

A similar study by Jiyao Sheng et al. investigated the role of mitochondria/lysosome interactions in HCC. Specifically, inhibition of PI3K/mTOR increased the sensitivity of hepatocellular carcinoma cells to cisplatin, by interfering with mitochondria/lysosome interactions [164]. Cisplatin triggered mitophagy and lysosomal biogenesis, leading to pharmacological resistance in HCC cells. The combination of cisplatin and the PI3K/mTOR inhibitor PKI-402 induced lysosomal membrane permeabilization and turned lysosomes from a protective role to a promoter of apoptosis. Strikingly, the authors demonstrated that this process was possible by the complete disruption of mitochondrial-lysosomal crosstalk [164]. This evidence suggests that targeting the autophagic machinery might represent a strategy to improve/rescue the metabolic asset of tumor cells and sensitize them to the existing or future therapeutical approaches.

Another and a recently explored feature of autophagy in cancer biology is its role in the regulation of lipid metabolism. Bhatt et al. proved that autophagy supported metabolic adaptations in LKB1-deficient KRas-driven lung cancer [165]. In fact, autophagy inhibition decreased recycling processes and limited amino acid metabolism via the TCA cycle, favoring fatty acid catabolism through β-oxidation and intracellular energetic crisis in lung cancer cells.

Andrejeva et al. also demonstrated that the synthesis of specific lipid species (e.g., phosphatidylcholine) was necessary for autophagosome biosynthesis in tumor cells [166]. During autophagy de novo choline phospholipid production and activation of phosphate cytidylyltransferase 1, choline, alpha (PCYT1A), the rate-limiting enzyme of phosphatidylcholine synthesis, occurs in the HCT116 human colorectal cancer cells. Strikingly, loss of PCYT1A activity results in altered autophagosome formation and maintenance in autophagic cancer cells. Notably, the authors also demonstrated that choline phospholipids of autophagic cells had longer-chain fatty acids, meaning that de novo choline phospholipid synthesis in tumor cells might contribute to generate a specific lipid asset for the formation of autophagosome [166]. 

Collectively, these works represent an important step forward in understanding the molecular basis of tumor cell biology and might pave the way for therapeutical strategies aimed at controlling autophagy, depending on the features of each tumor.

## 5. Conclusions

Mounting evidence indicates that mitochondria represent the Achille’s heel of the cell, in the transition from a normal to uncontrolled proliferation state, leading to cancer development. 

Tumor transformation, by altering several pathways and mechanisms described in this review, eludes the control systems of mitochondria, in order to produce or induce a metabolic switch, together with blunted apoptosis, both conferring growth advantages. The tethering between subcellular organelles has long been known, however only recently scientists turned their attention to finely investigate organelles’ interactions, both in physiology and disease, to unveil how they are affected. The changes induced by tumor transformation on mitochondria involve several pathways that lead to alterations of cell metabolism, and mitochondria dynamics that ultimately results in cell survival instead of death. Moreover, considering the complexity of organelle interactions, the high number of regulatory factors and molecules involved, it is conceivable that unknown molecular mechanisms still need to be unraveled. Multiple approaches were recently developed to investigate organelle cross-talk and interactions, ranging from genetic manipulation of structural or regulatory factors, to imaging techniques in live cells and metabolic profiling. Even more than in other research areas, a multipronged strategy is therefore essential to fully describe the phenotypic changes associated with organelles’ dynamics in cancer cells, the causative events, and the functional consequences. Research in the field greatly benefited from omics techniques and more achievements are expected in the future. Their application in “single-cell” mode would be extremely useful to disentangle phenotypic heterogeneity within the same tumor, which is one of the main obstacles in cancer biology.

## Figures and Tables

**Figure 1 cells-09-02519-f001:**
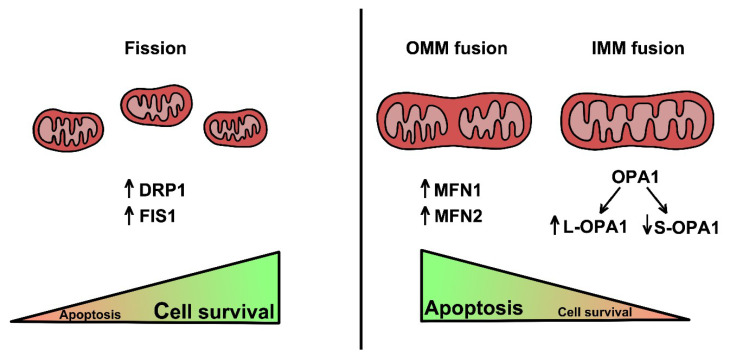
Mitochondrial fission is mainly mediated by DRP1 and FIS1 proteins, associated with increased aerobic glycolysis. This peculiar metabolic asset is typical of tumor cells to support their metabolic demands and their survival. On the other hand, elongation of mitochondria is a two-step event mediated by MFN proteins and OPA1. MFN1 and MFN2 participate in outer mitochondria membrane fusion, while one of OPA1 isoforms (L-OPA1) promotes IMM fusion. Conversely, another OPA1 isoform (S-OPA1) mediates to the opposite trend.

**Figure 2 cells-09-02519-f002:**
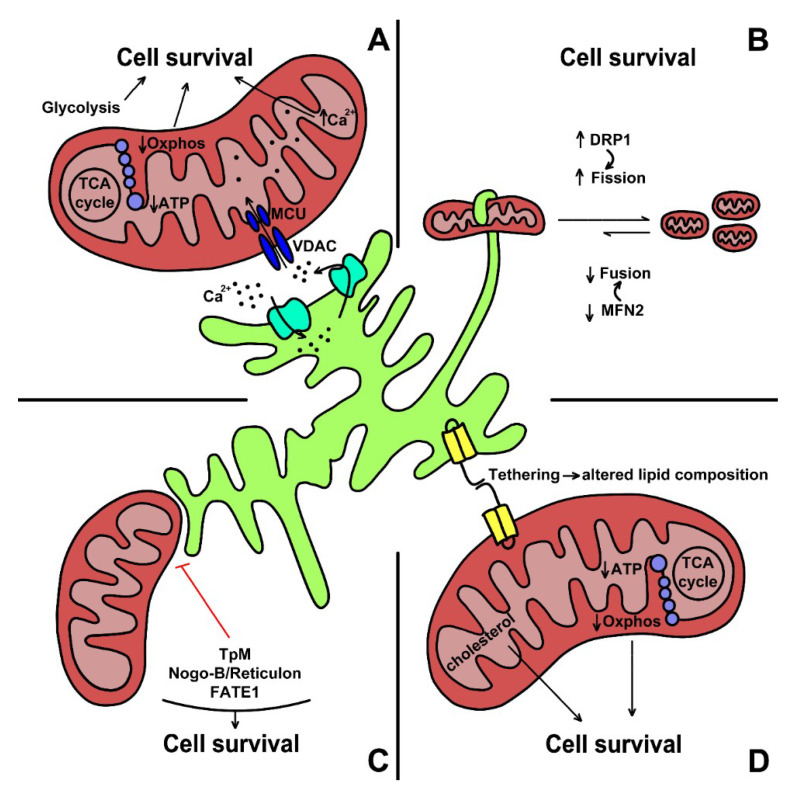
Main mechanisms involved in MAMs development and their role in tumor cell progression. (**A**). Calcium homeostasis is crucial to promote cell survival. A reduced flow of Ca^2+^ into mitochondria blunts Oxphos activity, thus, reducing ATP production. Moreover, Ca^2+^ induces the glycolytic pathway (Warburg effect), which eventually support cancer cells’ survival and proliferation. (**B**) Mitochondria fusion and fission programs are strictly controlled to be balanced in a physiological condition. However, during cancer progression, ER mediates mitochondria fission and shifts the equilibrium towards the fragmentation process. DRP1 and MFN2 are the main proteins involved in this context—the former enhances fission while the latter regulates mitochondrial fusion. (**C**) Tumor cells’ survival depends on the ER-mitochondria distance. Apoptosis happens as a consequence of their proximity, while a greater distance is associated with cell survival, as shown for Tpm, Nogo-B/Reticulon, and FATE1 proteins. (**D**) ER-mitochondrion tethering alters mitochondrial lipid composition. Specifically, their binding increases cholesterol levels, boosting up aerobic glycolysis and reducing Oxphos activity.

**Figure 3 cells-09-02519-f003:**
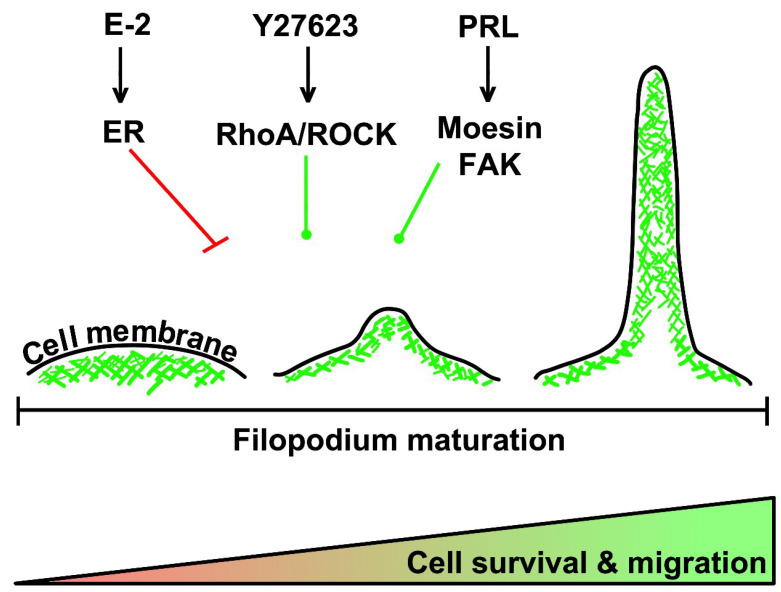
Membrane rearrangement is a pre-requisite for filopodium maturation, cancer cell migration, and metastasis. Different compounds can act on actin reorganization; for example, E2-treated cells display fewer protrusions and more cortical actin bundles, thus, blocking tumor cell migration. ROCK inhibitor (Y27623) causes a decrease in cortical actin bundles and concomitantly enhances protrusive activity, promoting cell migration and metastasis. Actin reorganization is also mediated by PRL, which controls moesin and FAK proteins, both involved in pseudopodia formation and cell migration.

**Table 1 cells-09-02519-t001:** Activities played by mitochondrial fusion and fission regulators in different kind of tumors.

Gene	Biological Function	Role in Cancer	Refs.
MFN1	Outer mitochondrial membrane fusion	MFN1 depletion led to: - epithelial-to-mesenchymal transition of hepatocellular carcinoma (HCC).	[40]
		- inhibition of mitochondrial fusion in human HCC and in vitro tumor organoids from cholangiocarcinoma, leading to reduced tumor formation.	[41]
		- ovarian cancer resistance to cisplatin.	[42]
MFN2	Outer mitochondrial membrane fusion	- MFN2/SIRT1-promoted mitophagy facilitating cell migration in gastric tumor.	[44]
		- controversial role of MFN2 in pancreatic cancer.	[45,46]
		- downregulation of MFN2 in bladder cancer	[47]
OPA1	Inner mitochondrial membrane fusion	- OPA1 depletion led to inhibition of mitochondrial fusion in human HCC and in vitro tumor organoids from cholangiocarcinoma, leading to reduced tumor formation.	[41]
		- increased OPA1 levels in ovarian cancer.	[53]
		- OPA1 mediates the antitumoral activity of Withaferin A in breast cancer cells.	[54]
		- OPA1 is required for tumor angiogenesis.	[55]
		- blunted OPA1 expression levels in invasive glioma.	[56]
		- OPA1 cleavage and mitochondrial fission impairment in neuroblastoma due to alterations in the function of KIF1Bβ/YME1L1/OPA1 axis.	[57]
DRP1	Mitochondrial fission	- DRP1 is upregulated in pancreatic cancer cell lines and its deletion conferred a significant survival advantage in a model of KRas-driven tumor.	[60,61]
		- DRP1 drives a metabolic rewiring in KRas mutant non-small-cell lung cancer and favors resistance to chemotherapy.	[62,63]
		- in lung adenocarcinoma, PINCH-1 inhibition led to increased DRP1 activity resulting in inhibition of cell proliferation.	[64]
		- antitumoral activity of DRP1 has also been proposed in an in vitro model of liver cancer	[65]
		- DRP1 decreased in malignant colon and lung cancer tissues, whereas no changes in DRP1 levels were observed in breast and prostate tumors.	[66,67,68]
		- DRP1 phosphorylation at Ser616 was also implicated in chloroquine- and isorhamnetin-mediated inhibition of triple-negative breast cancer cells growth and proliferation.	[69]
		- DRP1 mediates ovarian cancer resistance to cisplatin.	[42]
		- DRP1 is required for anti-tumor response of T-cells	[72]
FIS1	Mitochondrial fission	- FIS1 is overexpressed in the bone marrow of acute myeloid leukemia (AML) patients, and high FIS1 levels showed a significant negative impact on complete remission response after therapy. Human AML leukemia stem cells (LSCs) display high activity of AMPK/FIS1 axis, FIS1 depletion decreased mitophagy and increased cell differentiation of LSCs in AML.	[74]
		- the JNK-FIS1 axis is important for mitochondrial stress mediated by SIRT3 inhibition in tongue cancer.	[75]

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
