# Peer review of "“The Loss of Golden Touch”: Mitochondria-Organelle Interactions, Metabolism, and Cancer"

_cells, 2020, doi:10.3390/cells9112519_

Round 1
Reviewer 1 Report
This review gives a comprehension summarization of the role of mitochondria in the transition from a normal to uncontrolled proliferation state leading to cancer development. However, the minor revision will have an important impact in the area and the readers of the journal will have a better understanding of this manuscript.
Major comments:
- The introduction is too long. Authors provide a detailed introduction in the biochemical aspect of mitochondrial metabolism. I would suggest listing a subsection for this content, which makes the outline clearer. It has been known the shift in energy metabolism from oxidative phosphorylation to aerobic glycolysis in cancel cells. It will be important to provide more information regarding Warburg effects in this section.
- In the mitochondrial fusion and fission section, authors listed several related genes in cancer development. It would be meaningful if one summary table is included. The function of each genes, the relation with the cancer, the reported cancer type in the publications should be listed in the table. It would be easier for authors to review.
- All the figures are in low resolution, which need to be improved.
Minor comment:
- In line 103 and line 378, the same text “aerobic glycolysis (the so-called Warburg effect)” appeared twice. One could be eliminated.
Author Response
We thank the reviewers for their insightful comments. We have carefully received and addressed their suggestions to enhance our manuscript. We have also thoroughly revised the text to enhance the clarity and presentation of our review. As a result of these additions and changes, the paper is considerably improved, and we are indebted to the editor and the reviewers for the aid. We believe that we have addressed all concerns as detailed below:
Reviewer 1:
This review gives a comprehension summarization of the role of mitochondria in the transition from a normal to uncontrolled proliferation state leading to cancer development. However, the minor revision will have an important impact in the area and the readers of the journal will have a better understanding of this manuscript.
Major comments:
- The introduction is too long. Authors provide a detailed introduction in the biochemical aspect of mitochondrial metabolism. I would suggest listing a subsection for this content, which makes the outline clearer. It has been known the shift in energy metabolism from oxidative phosphorylation to aerobic glycolysis in cancel cells. It will be important to provide more information regarding Warburg effects in this section.
Response: we thank the reviewer for bringing up this important point, we now reshaped the previous version of introduction and re-named this part as “Overview of cancer cell metabolism”. In addition, we listed four different subsections to improve the flow of the review and better guide the reader. Moreover, we also included more information about the Warburg effect in the revised version of our manuscript.
- In the mitochondrial fusion and fission section, authors listed several related genes in cancer development. It would be meaningful if one summary table is included. The function of each genes, the relation with the cancer, the reported cancer type in the publications should be listed in the table. It would be easier for authors to review.
Response: as suggested by the reviewer, we added a table (table 1) summarizing the role of mitochondrial fission and fusion regulators in different kind of cancers with relative references.
- All the figures are in low resolution, which need to be improved.
Response: we now uploaded high resolution figures.
Minor comment:
- In line 103 and line 378, the same text “aerobic glycolysis (the so-called Warburg effect)” appeared twice. One could be eliminated.
Response: we amended the text as suggested by the reviewer.

Reviewer 2 Report
The manuscript by Audano et al., reviews current literature regarding mitochondria-organelle interactions.
They first revisit concepts of fission and fusion by describing functions of MFN, OPA, DRP1and FIS1.
They then go on and discuss interactions between mitochondria-ER (including Ca signalling), mitochondria-cytoskeleton and autophagy.
Overall, this is a nicely written review and I have no major comments.
Two small comments:
- Chapter 3.2 (autophagy) does somehow not perfectly fit under the overarching heading "inter-organellar communications" and is not ideally placed between Mito-ER (chapter 3.1) and mito-cytoskeleton (chapter 3.3) interaction. Maybe, the authors can highlight a bit more the relation of autophagy and mitochondria?
- resolution of the figures is not very good and the writing "figure" in the top left corner of each figure can be removed.
Author Response
We thank the reviewers for their insightful comments. We have carefully received and addressed their suggestions to enhance our manuscript. We have also thoroughly revised the text to enhance the clarity and presentation of our review. As a result of these additions and changes, the paper is considerably improved, and we are indebted to the editor and the reviewers for the aid. We believe that we have addressed all concerns as detailed below:
Reviewer 2:
The manuscript by Audano et al., reviews current literature regarding mitochondria-organelle interactions.
They first revisit concepts of fission and fusion by describing functions of MFN, OPA, DRP1and FIS1.
They then go on and discuss interactions between mitochondria-ER (including Ca signalling), mitochondria-cytoskeleton and autophagy.
Overall, this is a nicely written review and I have no major comments.
Two small comments:
- Chapter 3.2 (autophagy) does somehow not perfectly fit under the overarching heading "inter-organellar communications" and is not ideally placed between Mito-ER (chapter 3.1) and mito-cytoskeleton (chapter 3.3) interaction. Maybe, the authors can highlight a bit more the relation of autophagy and mitochondria?
Response: we thank the reviewer, and we agree with her/him suggestion. Therefore, we move the autophagy as a stand-alone part that in the revised version of the manuscript is chapter 4. Nevertheless, we also implemented the relationship between autophagy and mitochondria
- resolution of the figures is not very good and the writing "figure" in the top left corner of each figure can be removed.
Response: we now uploaded high resolution figures and removed the name “figure” from the top left corner.
